# Operational, Economic, and Environmental Assessment of an Agricultural Robot in Seeding and Weeding Operations

**Mahdi Vahdanjoo** [1,*], **René Gislum** [1] **and Claus Aage Grøn Sørensen** [2]

1 Department of Agroecology, Aarhus University, Forsøgsvej 1, 4200 Slagelse, Denmark
2 Department of Electrical and Computer Engineering, Aarhus University, Finlandsgade 22, 8200 Aarhus, Denmark
* Correspondence: mahdi.vahdanjoo@agro.au.dk

**Abstract:** The development of robotic-based agricultural machinery systems has significantly increased in recent years. Many autonomous systems have not yet been measured based on sustainability and economic performances, even though automation is regarded as an opportunity to increase safety, dependability, productivity, and efficiency. The operational aspect, economic viability, and environmental impact of replacing conventional machinery with robotized alternatives are the primary focus of this study. The robot considered in this research is designed for extensive fieldwork, where PTO and external hydraulics are required. This robot is equipped with two 75 (hp) Kubota diesel engines with a total engine gross power of up to 144 (hp). Both robotic system and conventional machinery were described, and different scenarios were used to examine various operational and environmental indicators, as well as individual cost elements, considering various field sizes and working widths of implements used in seeding and weeding operations. The findings demonstrate that the robotic system outperforms conventional machinery in terms of operational efficiency by as much as 9%. However, the effective field capacity comparison reveals that the conventional system has a field capacity that is up to 3.6 times greater than that of the robotic system. Additionally, the total cost per hour of the robotic system is up to 57% lower than that of the conventional system. The robotic system can save up to 63.3% of fuel during operation, resulting in the same percentage reduction in $CO_2$ emissions as the conventional system, according to a comparison of fuel consumption.

**Keywords:** economic assessment; environmental impact; operational management; precision agriculture; robotic system; sustainability

## 1. Introduction

In recent years, the introduction of agricultural machinery systems based on robotic applications has increased significantly [1–3]. The main reason behind the automation of agricultural processes is the need for more efficiency. This can be achieved by reducing the operational time, reducing the energy required to perform machine operations, and increasing the crop yield. Moreover, recent advances in mechanical engineering, sensory perception, computing, and human interfacing are laying the foundation for the introduction and application of autonomous robotic systems.

Automation is seen as a chance to boost productivity, safety, reliability, and efficiency. However, the sustainable environmental and economic performances of many autonomous systems have yet to be measured [4]. There have been few studies on the economic and operational feasibility of using autonomous machines in agriculture. However, the economic, operational, and environmental aspects of robotic systems have not been the subject of comprehensive assessments [5]. Earlier studies such as Goense (2003) [6] indicate the positive feasibility of autonomous vehicles when utilized for up to 23 (h/day) by setting up scenarios for balanced cropping practices and cropping rotations for grains.

Have (2004) [7] showed the effects of automation on machinery sizes and costs for soil tillage and crop seeding operations including all machinery costs as well as timeliness costs. Results showed 20% higher investment, 80% lower labour requirement, and double the working hours, as well as showing that changing to autonomous vehicles would lower the machinery sizes and investment to 50–60% and the overall costs to 65%. Sørensen et al. (2005) [8] assessed the feasibility of a plant-nursing robot for weeding operations and found that profitability gains ranging from 20 to 50% are achievable through targeted applications. Specific studies for robotic seeding in sugar beets indicate a 7–10% cost reduction [9]. Lightweight robotic systems are expected to reduce soil compaction and its adverse effects on soil properties such as field readiness and soil workability which can cause poor drainage and increase surface runoff [10]. As a conclusion, indications are that applying an autonomous system would reduce the size, initial cost of machinery, and operation cost [11]. There are different types of agricultural robots that can automate seeding or weeding operations. Dahlia 4.3 is a lightweight and fully automatic robot that uses solar power as source of energy [12]. Farming-GT is another agricultural robot that is fully electrical and lightweight [13]. Farm Droid FD20 is another fully automatic and lightweight robot that is solar-powered [14]. Naio-OZ is another fully electrical and lightweight robot that can automate sowing and weeding operations [15]. For this research, the Agrointelli-Robotti [16] was used, which is a powerful field robot that uses diesel as a source of energy to accomplish power-intensive operations.

Seeding as the basic operation in crop production has a major effect on the productivity and environmental footprint of agriculture. Recent studies have shown a growing interest in the application of robotic systems in precision seeding operations. Improper tillage and seeding practices can cause extensive yield losses [9]. Bhimanpallewar & Narasingarao (2020) [17] presented an automatic seeding and fertilizer micro-dosing robot for farmers to perform precision farming. The designed robot can carry out the seeding and micro-dose fertilizing operations based on the type of seeds and plants, including quantifying the number of seeds and the amount of applied fertilizer. Moreover, the performance of this robot in terms of working time and energy consumption was analyzed in each soil profile for each type of mentioned seeds. However, no comprehensive evaluation of capacity and efficiency, and comparison with traditional methods were carried out. Neha S. Naik et al. (2016) [18] proposed a prototype of an autonomous agriculture robot that is specifically designed for seed operation. The designed robot is a four-wheeled vehicle that can perform efficient seed sowing at optimal depth and distances between crops and their rows. However, no analysis of different performance parameters for the designed robot was carried out. Sunitha et al. (2017) [19] designed an agricultural robot for plowing and seed-sowing operations in the field however, the detailed performance of the robot was not assessed.

There are three approaches to weeding: mechanical weeding, chemical weeding, or a combination of them. In mechanical weeding, the weed plants are removed by uprooting, cutting, and/or flaming. In chemical weeding, the weeds are removed by spraying herbicides on the plants. Recent studies have shown that weeding robots have received significant attention from researchers. Gonzalez-de-Soto et al. (2016) [20] presented an agricultural robotic solution for precise spraying. The designed system consists of an autonomous mobile robot based on a modified commercial tractor (for example equipped with high-tech perception and actuation systems), a real-time machine vision system that can detect weeds, and a rapid-response spraying system. Laboratory characterization and field tests revealed that the proposed system was reliable and can treat approximately 99.5% of the detected weeds while achieving significant herbicide savings. However, the author did not evaluate all the performance parameters of this robot from an agronomic, operational, and sustainability point of view, as well as not comparing conventional methods. Berenstein & Edan (2017) [21], presented a human–robot collaboration for site-specific spraying. The article provides details on the robotic platform design, the human–robot collaboration framework, and the tools used for the robotic sprayer to collaborate with a

remote human operator for target detection and spraying. The field experiment proved the feasibility of human–robot collaboration for spraying specific targets. Moreover, the proposed system can help to remove humans from hazardous pesticide environments as well as to reduce the use of pesticides by up to 50% by minimizing the quantity of sprayed material while maintaining the crop yield. In the presented work, only a limited performance evaluation was carried out.

This paper proposes a targeted methodology for the evaluation of operational, economic, and environmental aspects of a robotic system in arable farming. This methodology is based on analyzing the basic unit tasks and processes for estimating the operational performance of the robotic system and traditional agricultural machinery systems.

In the following sections, a conventional machinery system and a robotic system are described and analyzed through varying scenarios, by considering different field sizes, and different working widths for the implements in seeding and weeding operations, followed by a breakdown of all costs and environmental impact assessment. The hypothesis is that this methodology can assess the operational aspect, economic feasibility, and environmental impact of replacing conventional practices with a respective robotized alternative.

## 2. Materials and Methods

### 2.1. The Robot

The analyzed robot (Figure 1) performs site-specific agricultural operations including seedbed preparation, seeding, hoeing, weeding, harrowing, soil sampling, spraying, and mowing. Version 150D of this robot is designed for extensive fieldwork, where PTO and external hydraulics are required for example in seeding operations. The mentioned version of the robot is equipped with two 75 (hp) Kubota diesel engines (total engines gross power up to 144 (hp)). One engine is dedicated to propulsion, the other runs the PTO and external hydraulic system. The forward speed of the robot in autonomous mode is up to 5 (km/h) and the high speed in manual mode is up to 10 (km/h). The approximate weight of the robot is 3100 (kg), and it has 2-wheel steering (able to do zero turns). The robot navigates precisely with Real-Time kinematic (RTK) GPS technology that has an accuracy of 2 (cm). The logging data collected from the control center of the robot were analyzed to measure the in-field operations of the robot [16].

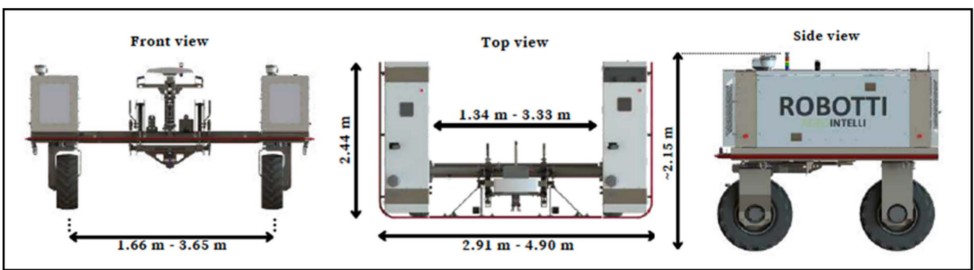

**Figure 1.** The robotic system.

### 2.2. Monitoring, Measuring, and Analyzing of Robot's In-Field Operations

By using the GPS antenna and sensors, it is possible to measure the in-field performance of the robot. The motions of the robot are decomposed into different task (time/distance) elements. The generated data can be aggregated into useful information for determining the fieldwork patterns, turning types, and evaluation of performance parameters such as field capacity, field efficiency, and so on. These data can be used for analyzing the current coverage method to identify the potential efficiency of the applied robotic system. This includes a comparison of robot vs. conventional method in terms of performance parameters. Figure 2 is the plotted GPS coordinates related to the location of the robot. During the operation, the GPS antenna of the robot sends the coordinates of the robot's location and based on the logging data from the robot (e.g., the mode of the robot:

manual driving, automatic and working, automatic and not working) we can plot with different colors to distinguish different task elements of the robot.

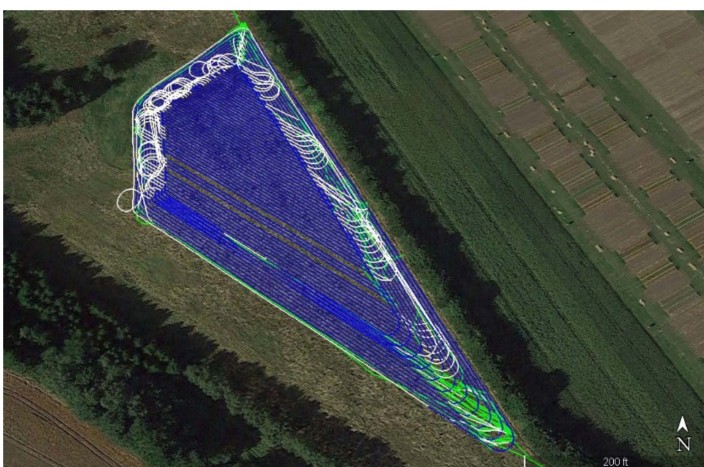

**Figure 2.** An example of analysis for the robot's in-field operation. "Manual driving": When the robot drives manually by the robot's operator (Green color). "Automatic and working": When the robot automatically performs the operation in the field (Blue color). "Automatic and not working": When the robot automatically moves but not performing the operation for example when it turns in the headland part (White color).

### 2.3. Operational Factors

In this study, specific operational indicators were considered, including field efficiency, effective field capacity, theoretical field capacity, energy consumption rate, and power requirements of the machine. These factors are important for evaluating the capacity and efficiency of an agricultural machine [22].

### 2.3.1. Field Efficiency

It is a comparison of the actual amount of work done by a machine compared to what it would do without any loss of time or capacity.

$$Field\ Efficiency\ (\%) = \frac{100 \times T_p}{T_t} \tag{1}$$

where $T_p$ = productive time (actual working) and, $T_t$ = total time (considering the static and idle time).

### 2.3.2. Effective Field Capacity

The effective field capacity is determined by considering the working speed, the working width of the implement, and the machine's field efficiency.

$$Effective\ Field\ Capacity\ (ha/h) = \frac{S \times W \times Field\ Efficiency}{C} \tag{2}$$

where $S$ = working speed (km/h), $W$ = working width of implement (m), and a constant $C$ = 10.

### 2.3.3. Theoretical Field Capacity

The theoretical field capacity is determined by noting the theoretical work speed and the machine's theoretical working width.

$$Theoretical\ Field\ Capacity\ (ha/h) = V_t \times W_t \times K \tag{3}$$

where $V_t$ = theoretical working speed, $W_t$ = theoretical working width, and a constant $K = 0.1$.

*2.4. Economic Factors*

The total cost of an agricultural machine can be calculated as the summation of two main parts, the ownership cost, and the operational cost [3].

2.4.1. Ownership Costs

The costs of ownership include depreciation, interest cost, taxes, insurance, and housing facilities [3]. Depreciation is a type of expense that results from the obsolescence and age of a machine [23]. The degree of mechanical wear may cause a reduction in the initial value of a particular machine. Sometimes a major design change or emerging new technology can make an older machine suddenly obsolete, causing a sharp drop in its residual value [23]. However, the economic life and total hours of usage are usually considered as the main factors for calculating the residual value of a machine. To estimate the annual depreciation, the machine's age, and the salvage value at the end of the economic life must be specified. The economic life of a machine is the number of years for which costs should be estimated. The salvage value (SV) is assumed to be 10% of the machine's purchase price (PP) [3]

$$c_d = \frac{PP - SV}{Age} \qquad (4)$$

where $PP$ = purchase price, $SV$ = Salvage value, and $Age$ = economic life.

If the operator borrows money to purchase a machine, the lender sets the interest rate to charge. However, if the farmer uses his own capital, the rate will depend on the opportunity cost for that capital elsewhere in the farm business. If only part of the money is borrowed, an average of the two rates should be used. The average annual interest charge is computed by subtracting the trade-in or salvage value from the purchase price, multiplying this difference by the rate of interest, and dividing by 2 [23]. In this research, it is assumed that the interest rate $i$ is equal to 9 percent.

$$Interest = \frac{PP - SV}{2} \times i \qquad (5)$$

where $i$ = interest rate.

The cost of taxes and insurance is usually much lower than depreciation and interest, but it should be considered. In this research, a cost estimate equal to 1.0 percent of the purchase price is considered as the cost of taxes and insurance [24].

Providing shelter, tools, and maintenance equipment leads to fewer repairs in the field and less damage to mechanical parts. In this research, an estimated charge of 1.0 percent of the purchase price is considered for housing costs [24]. The estimated costs of depreciation, interest, taxes, insurance, and housing are added together to find the total ownership cost.

2.4.2. Operational Costs

The operating cost of a system is the sum of maintenance costs, energy costs (i.e., fuel or electrical energy consumption), lubrication, labor costs, and farm-to-field transportation costs. For the calculation of the repair and maintenance cost for the conventional machinery system, the ASABE standards estimation process is applied [25]

$$C_{rm} = RF1 \times \left( \frac{h}{1000} \right)^{RF2} \times PP \qquad (6)$$

where $RF1$ and $RF2$ are the repair and maintenance factors and $h$, expressed in hours, stands for the accumulated working hours of the machinery.

The performance rate of agricultural machines depends on the speed that can be achieved and the optimal use of time [25]. Field speeds may be limited by heavy yields,

rough soil, and adequacy of operator control [25]. Small or irregularly shaped fields, heavy yields, and high-capacity machines may cause a substantial reduction in field efficiency [25]. Typical speeds and field efficiency for seeding operation are presented in Table 1.

**Table 1.** Operational data and repair factors for conventional machine types [25].

| Machine | Field Efficiency | | Field Speed | | Estimated Life | Total Life R&M Cost | Repair Factors | |
|---|---|---|---|---|---|---|---|---|
| | Range % | Typical % | Range (km/h) | Typical (km/h) | H | % of List Price | RF1 | RF2 |
| Grain drill | 55–80 | 70 | 6.5–11.0 | 8.0 | 1500 | 75 | 0.32 | 2.1 |
| Tractors (4-wheel drive & crawler) | | | | | 16,000 | 80 | 0.003 | 2.0 |

For the case of conventional machinery, the prediction of the energy cost, that is the engine fuel consumption during the operation, was based on the specific volumetric fuel consumption approach as it is described in Table 2.

**Table 2.** Average fuel consumption for different modes of tractor operation [26].

| Average Fuel Consumption (John Deere 7250 R) | | | |
|---|---|---|---|
| Working width sowing machine (m) | 3 | 4 | 6 |
| Power demand (kW) | 90 | 105 | 155 |
| **Mode** | **Consumption (L/h)** | | |
| Working mode (sowing) | 13.82 | 15.64 | 23.08 |
| Transportation mode | 9.61 | 11.21 | 16.55 |
| Static mode | 4.30 | 4.30 | 4.30 |

In the standard ASABE and "Farm power and machinery management" [27], the required energy for pulling a Rod weeder is in the range of 1.5–4.8 (kW h/ha). Therefore, based on the "Nebraska OECD Tractor Test 2085–Summary 934, John Deere 7250r Diesel, E23 Transmission" [28], the amount of fuel consumption for the tractor in the weeding operation was considered 9.43 (L/h).

According to the standard ASAE D497.4 FEB03 [29], the amount of oil consumption in (L/h) can be estimated for tractor and robot based on their maximum PTOP. For the PTOP between 134–200, the amount of oil consumption for a diesel engine is 0.111 (L/h) and for PTOP over 200, the amount of oil consumption is equal to 0.135 (L/h).

In a robotic system, although the system performs autonomously, manual labor may require supervising the so-called residual tasks (such as system advancements, reconfigurations, and potential safety monitoring) [3]. In this research, for the labor cost of the robotic system, half of the hourly wage of the conventional system was considered. Finally, we could also consider the transportation cost for the robotic system in the case where the distance to the field is significant.

## 2.5. Environmental Factors ($CO_2$ Emission)

The amount of emission $CO_2$ gas during the seeding and weeding operation can be estimated based on the amount of consumed fuel by the tractor or robot. According to other studies [30–32], consuming 100 L of diesel fuel emits 275–376 (kg) of $CO_2$. Therefore, in this study, it is assumed that the consumption of 1 L of diesel fuel during the operation produces 2.75 (kg) of $CO_2$ gas related to the greenhouse effect.

*2.6. Comparing the Performance Parameters of a Robotic System with Conventional Agricultural Machinery*

2.6.1. Comparing the Draft Force Required to Pull an Implementation

The objective of this test was to find out how much energy it takes for an implement to be pulled by the robot. The results of this test were used to compare the energy requirements of the robot with the conventional machinery. An eight-directional string gauge device demonstrated in Figure 3 was mounted on the robot, and thereafter a Kongskilder seeder was mounted onto the string gauge device.

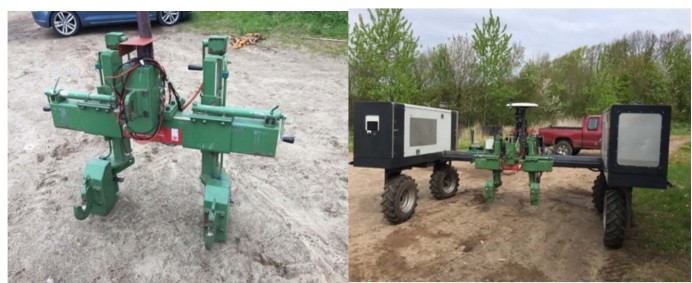

**Figure 3.** Mounted string gauge on the robot.

Two different implementations were mounted on the robot in this pulling test. Figures 4 and 5 demonstrate the applied seeder and weeder machines. The specification of these machines is listed as follow:

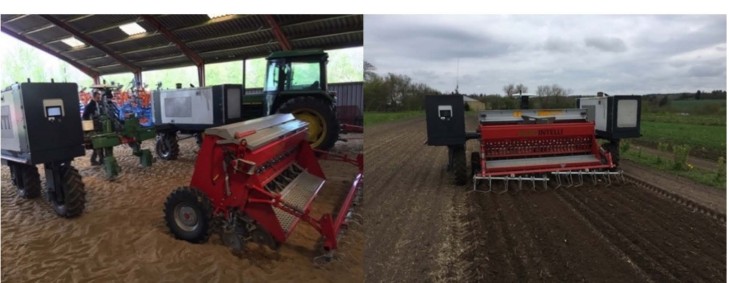

**Figure 4.** Grain drill seeder, 3 (m)/25 rows, 5 (km/h), 4 (cm) depth. The seeder was filled with weights in the center to represent the weight of the seed. Total weight 210 (kg) (4 × 30 (kg) + 2 × 45 (kg)).

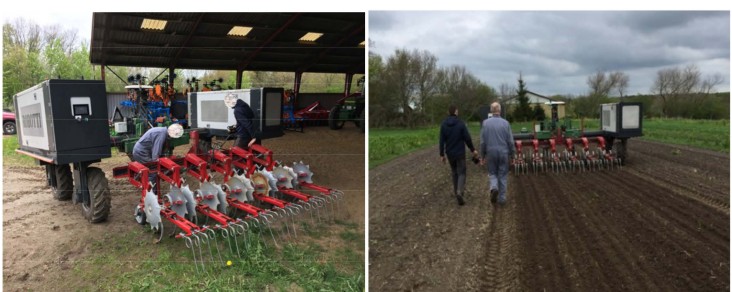

**Figure 5.** Mechanical weeder (Rod weeder), 3 (m), 5 (km/h), 2 (cm) depth.

The test field was in Denmark and the robot drove the planned route automatically. Only four tracks were considered for this test. The length of the tracks is T1 = 131 (m), T2 = 141 (m), T3 = 170 (m), and T4 = 189 (m). Figure 6 demonstrates the test field.

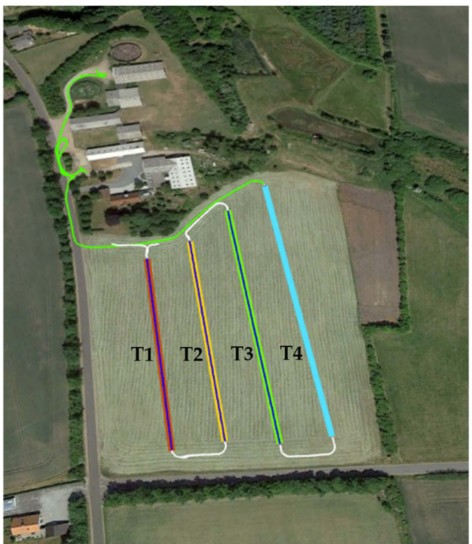

**Figure 6.** The sample field was used for the draft force test. T1, . . . , T4 represent the field work tracks.

The draft force (N) required to pull the implementation (in the horizontal direction) can be estimated based on Equation (7). The soil and machine parameters for the weeder and seeder were presented in Table 3.

$$Draft\ (N) = F_i \left[ A + (B \times S) + \left( C \times S^2 \right) \right] WT \tag{7}$$

$F_i$ = dimensionless soil texture adjustment parameter; $i$ = 1 for fine, 2 for medium, and 3 for coarse-textured soils; $A$, $B$, and $C$ = machine-specific parameters; $S$ = field speed (km/h); $W$ = machine width (m); $T$ = tillage depth (cm) for major tools. The value of $T$ for minor tillage tools and seeding implements is equal to 1 (dimensionless).

**Table 3.** Machine and soil parameters for rod weeder and grain drill [25].

| Implement | Width | Machine Parameters | | | Soil Parameters | | | Range ±% |
|---|---|---|---|---|---|---|---|---|
| | | A | B | C | F₁ | F₂ | F₃ | |
| | | | | Major Tillage Tools | | | | |
| Rod Weeder | m | 210 | 10.7 | 0.0 | 1.0 | 0.85 | 0.65 | 25 |
| | | | | Minor Tillage Tools | | | | |
| Grain Drill | rows | 720 | 0.0 | 0.0 | 1.0 | 0.92 | 0.79 | 35 |

### 2.6.2. Case Study Description

A case study is presented to show how to apply the methodological approach in this study. The purpose of the paper is to evaluate the operational, economical, and environmental aspects of the robotic system and compare them with conventional machinery. Different scenarios were defined based on the size of the fields (less than 1 (ha), between 1 to 10 (ha), and more than 10 (ha)), type of operations (seeding and weeding), and different working widths for the conventional machinery.

Table 4 shows how to calculate the ownership and operational costs for the robotic and conventional systems in seeding operations. As an example, the purchase price of the robot with seeding attachment sets to EUR 144,500, then the salvage value is 10% of the purchase price which is equal to EUR 14,450. The economic life of the robot is assumed to be 6000 h, equal to 10 years. The average interest rate is assumed to be equal to 9%. Depreciation can be evaluated based on the formula (4). The amount of interest can be calculated by the formula (5). The cost of insurance and taxes as well as the cost of housing assumed to be 1% of the purchase price. Then the ownership cost is equal to the summation of depreciation,

interest, insurance and taxes, and housing costs. When the field is too far, a trailer is going to use for the transportation of the robot to the field. In this case study, the distance to the fields is short and the transportation cost was not considered.

**Table 4.** The calculation of ownership costs for robotic and conventional machinery in seeding operation.

| | | Robot-Working Width 3 [m] | Tractor-Working Width 3 [m] | Tractor-Working Width 4 [m] | Tractor-Working Width 6 [m] |
|---|---|---|---|---|---|
| Ownership costs parameters | Purchase price (EUR) | 144,500 | 269,200 | 273,200 | 280,700 |
| | Salvage value (EUR) | 14,450 | 26,920 | 27,320 | 28,070 |
| | Economical life (years) | 10 | 15 | 15 | 15 |
| | Average interest rate (%) | 9 | 9 | 9 | 9 |
| | Depreciation (EUR/year) | 13,005 | 16,152 | 16,392 | 16,842 |
| | Interest (EUR/year) | 5852 | 10,903 | 11,065 | 11,368 |
| | Insurance and Taxes (EUR/year) | 1445 | 2692 | 2732 | 2807 |
| | Housing (EUR/year) | 1445 | 2692 | 2732 | 2807 |
| Ownership costs (EUR/year) | | 21,747 | 32,439 | 32,921 | 33,824 |
| Economical life (h) | | 6000 | 16,000 | 16,000 | 16,000 |
| Ownership costs (EUR/h) | | 3.62 | 2.03 | 2.06 | 2.11 |

In the case of conventional machinery, a John Deere tractor model (7R250) was considered as an example to show how to calculate the total cost for conventional machinery. For example, in the case of "tractor-working width 3 [m]" by setting the purchase price of the tractor with seeding attachment equal to 269,200 (EUR), then the salvage value is 10% of that, which is equal to EUR 26,920. The economic life of the tractor is assumed to be 16,000 h, equal to 15 years (each year is 1067 (h)) [33]. The average interest rate is assumed to be equal to 9%. The depreciation value and the amount of interest can be calculated by Equations (4) and (5) respectively. The cost of insurance and taxes as well as the cost of housing assumed to be 1% of the purchase price. Then the ownership cost is equal to the summation of depreciation, interest, insurance and taxes, and housing costs.

Table 5 shows how to calculate the ownership and operational costs for the robotic and conventional systems in weeding operations. The same procedure as above was applied by considering the price of weeding attachments in each case.

**Table 5.** The calculation of ownership costs for robotic and conventional machinery in weeding operation.

| | | Robot-Working Width 2.4 [m] | Tractor-Working Width 2.4 [m] | Tractor-Working Width 4 [m] | Tractor-Working Width 6 [m] |
|---|---|---|---|---|---|
| Ownership costs parameters | Purchase price (EUR) | 144,200 | 268,900 | 275,700 | 284,600 |
| | Salvage value (EUR) | 14,420 | 26,890 | 27,570 | 28,460 |
| | Economical life (years) | 10 | 15 | 15 | 15 |
| | Average interest rate (%) | 9 | 9 | 9 | 9 |
| | Depreciation (EUR/year) | 12,978 | 16,134 | 16,542 | 17,076 |
| | Interest (EUR/year) | 5840 | 10,890 | 11,166 | 11,526 |
| | Insurance & Taxes (EUR/year) | 1442 | 2689 | 2757 | 2846 |
| | Housing (EUR/year) | 1442 | 2689 | 2757 | 2846 |
| Ownership costs (EUR/year) | | 21,702 | 32,402 | 33,222 | 34,294 |
| Economical life (h) | | 6000 | 16,000 | 16,000 | 16,000 |
| Ownership costs (EUR/h) | | 3.62 | 2.03 | 2.08 | 2.14 |

## 3. Results and Discussion

### 3.1. The Result of Draft Force Test

The average of the horizontal force during the seeding operation was plotted in Figure 7 and the required energy was calculated as follows:

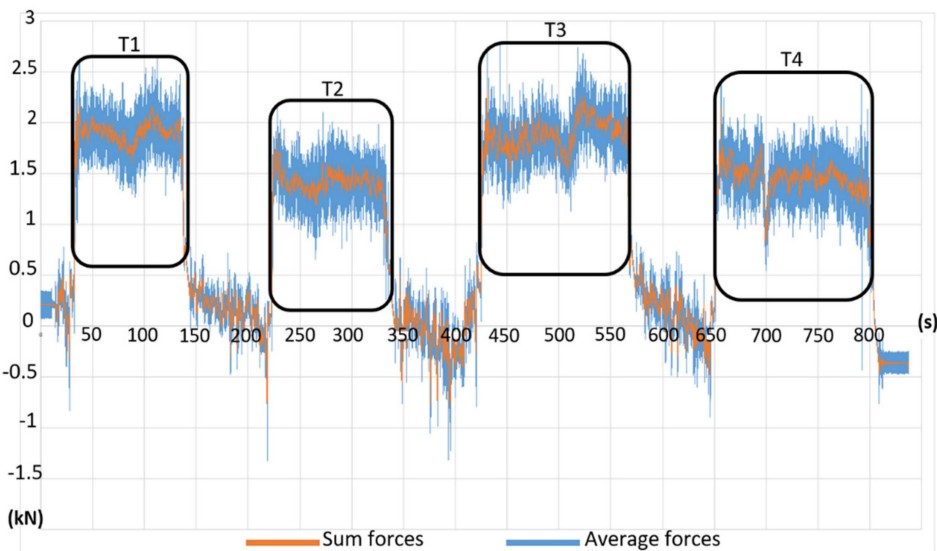

**Figure 7.** Plotted average horizontal force during the seeding operation. Vertical axis represents the forces (kN) and horizontal axis shows the time (s).

The required average power for covering each track can be calculated based on this formula, Power = Force × Distance/Time, therefore, for T1 the amount of power is equal to 2.223 (kW). To calculate the amount of (kW h), the following formula was used: kW hr = Power (kW) × Time (h). For example, in Seeding/T1, the (kW h) can be calculated as follow: kW hr = 2.223 × (109/3600) = 0.067 (kW h). The amount of covered area for each track can be calculated as follow: Work area = working width × track length. For example, in Seeding/T1: Work area = 3 (m) × 131 (m) = 393 (m$^2$) = 0.0393 (ha) (The working width of the implementations was 3 m). Finally, the amount of kW h/ha for Seeding/T1 can be calculated as follows: kW h/ha = 0.067 (kW h) × 1/0.0393(ha) = 1.7 (kW h/ha). Table 6 shows the result of the draft force test in the seeding operation.

**Table 6.** Calculation of required energy for pulling the seeder.

| Calculation of Required Energy for Pulling the Seeder | | | | | |
|---|---|---|---|---|---|
| | **Seeding/T1** | **Seeding/T2** | **Seeding/T3** | **Seeding/T4** | **Average** |
| The average horizontal force (kN) | 1.85 | 1.39 | 1.87 | 1.44 | 1.64 |
| Travel time (s) | 109 | 114 | 143 | 150 | 129 |
| Traveled distance (track length) (m) | 131 | 141 | 170 | 189 | 157.75 |
| kW h | 0.067 | 0.055 | 0.088 | 0.076 | 0.072 |
| kW h/ha | 1.70 | 1.30 | 1.73 | 1.33 | 1.52 |

Figure 7 shows the average horizontal force during the seeding operation. $T_i$, $i = 1, \ldots 4$ represents the track (working area) ID. The plotted forces in the boxes related to the horizontal force during the automatic performance of robot. The other horizontal forces related to the driving or turning in the headland part.

The same calculation as the seeder was considered for the mechanical weeding operation. The average horizontal force during the weeding operation was plotted in Figure 8 and the results were summarized in Table 7.

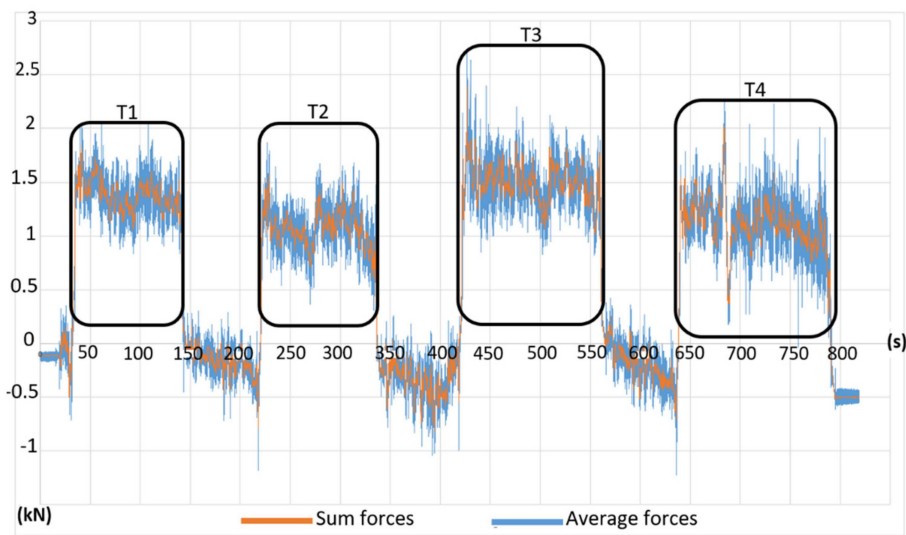

**Figure 8.** Plotted average of horizontal force (kN) during the weeding operation. Vertical axis represents the forces (kN) and horizontal axis shows the time (s).

**Table 7.** The results of the calculation of required energy for the weeding operation.

| Calculation of Required Energy for Pulling the Weeder | | | | | |
| --- | --- | --- | --- | --- | --- |
| | **Weeder/T1** | **Weeder/T2** | **Weeder/T3** | **Weeder/T4** | **Average** |
| The average horizontal force (kN) | 1.38 | 1.07 | 1.50 | 1.12 | 1.27 |
| Travel time (s) | 106 | 115 | 139 | 148 | 127 |
| Traveled distance (track length) (m) | 131 | 141 | 170 | 189 | 157.75 |
| kW h | 0.050 | 0.042 | 0.071 | 0.059 | 0.056 |
| kW h/ha | 1.28 | 0.99 | 1.39 | 1.04 | 1.18 |

Like Figure 7, Figure 8 also shows the average horizontal force during the weeding operation.

Based on the standard ASABE [25,27], we can see that for example for seeder (Grain drill) the required (kW h/ha) for pulling the implement is in this range (1.1–3.9) and for weeder (Rod weeder) is (1.5–4.8). The comparison shows that the required energy to pull the implementation with the robot is in the lower range than the required energy from the tractor.

### 3.2. Seeding Operation

In the first scenario, a test field (Figure 9) with a size of 0.56 (ha) (less than 1 hectare) was selected for the seeding operation. The driving was connected to route planning and the operation was executed automatically. The duration of operation was 67 min, the amount of automatic working time was 36 min, and the rest was automatic turning time.

Table 8 shows the result of in-field analysis for the robotic system and conventional machinery. The field efficiency and effective field capacity can be calculated based on formulas 1 and 2 respectively. The amount of fuel consumption for the tractor in different modes can be evaluated based on the data from Table 2. The amount of oil consumption for the robot can be calculated by assuming the consumption rate equal to 0.111 (L/h) and for the tractor, it can be estimated by assuming the consumption rate equal to 0.135 (L/h) [27]. The energy cost can be calculated by multiplying the fuel consumption by the fuel cost. The lubrication cost can be calculated in the same way as the energy cost. The price of diesel and lubricant is assumed to be EUR 2 and EUR 0.3 respectively. The labour cost for the robot is assumed to be EUR 14.8/h and for the tractor assumed to be double this amount and equal to EUR 28.6/h. The repair and maintenance cost for the robot is equal to EUR 7009 per year and by dividing it by 600 h, it is equal to EUR 11.7/h. The repair and maintenance cost for the tractor can be calculated based on the information from Table 1.

Then the operational cost is the summation of all the previous costs (energy, lubrication, labour, repair, and maintenance). In the end, if the ownership cost is divided by 6000 h for the robot and 16,000 h for the tractor, then it is going to have the same unit as operational cost (EUR/h) and the summation of operational and ownership costs will show the total cost. The amount of $CO_2$ emission can be estimated by multiplying 2.75 (kg $CO_2$) by the amount of fuel consumption.

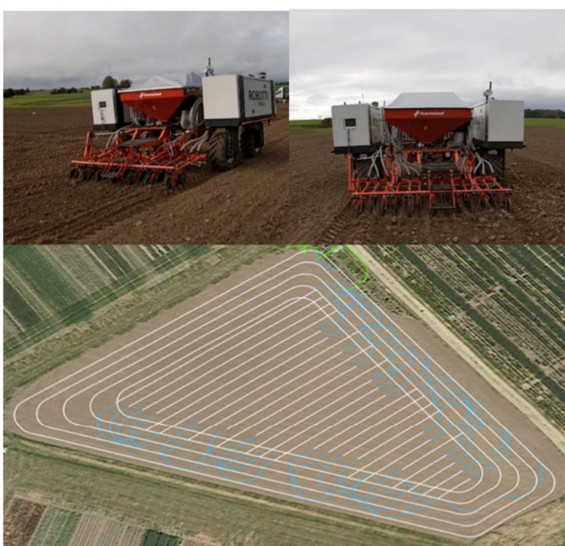

**Figure 9.** Seeding operation with the robot in a small triangle field.

**Table 8.** The results of in-field operation for the first scenario for the robot and the tractor with different working widths.

|  | Robot-Working Width 3 [m] | Tractor-Working Width 3 [m] | Tractor-Working Width 4 [m] | Tractor-Working Width 6 [m] |
|---|---|---|---|---|
| Field working time [min] | 36 | 15.4 | 11.8 | 8 |
| Non-working time [min] | 31 | 13.7 | 10.6 | 7.12 |
| Static time [min] | 0 | 0.95 | 0.34 | 0.34 |
| Total time [min] | 67 | 30.05 | 22.74 | 15.46 |
| Field efficiency [%] | 53.7 | 51.2 | 51.9 | 51.7 |
| Effective field capacity [ha/h] | 0.81 | 1.36 | 1.84 | 2.92 |
| Fuel consumption [L] | 3.5 | 5.82 | 5.12 | 5.06 |
| Oil consumption [L] | 0.12 | 0.07 | 0.05 | 0.03 |
| Energy cost [EUR] | 7 | 11.64 | 10.24 | 10.12 |
| Lubrication cost [EUR] | 0.04 | 0.02 | 0.015 | 0.01 |
| Labour cost [EUR] | 16.53 | 14.82 | 11.22 | 7.63 |
| Repair and Maintenance [EUR] | 13.1 | 6.76 | 5.2 | 3.62 |
| Operational costs [EUR] | 36.67 | 33.24 | 26.68 | 21.38 |
| Operational costs [EUR/h] | 32.84 | 66.37 | 70.4 | 83 |
| Ownership costs [EUR/h] | 3.62 | 2.03 | 2.06 | 2.11 |
| Total cost per hour [EUR/h] | 36.46 | 68.4 | 72.46 | 85.11 |
| Total cost [EUR] | 40.71 | 34.26 | 27.46 | 21.93 |
| $CO_2$ emission [kg] | 9.63 | 16 | 14.08 | 13.92 |

Figures 10 and 11 demonstrate the comparison of operational factors between the robot and the tractor. The results show that the robot has slightly better field efficiency than the tractor. However, the tractor has a higher effective field capacity than the robot.

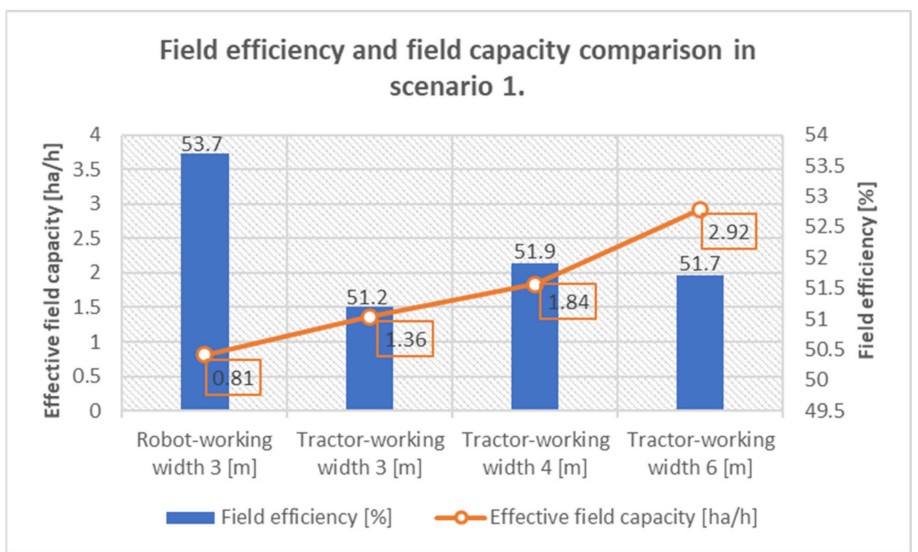

**Figure 10.** Comparison of field efficiency and field capacity for robot and tractor in scenario 1.

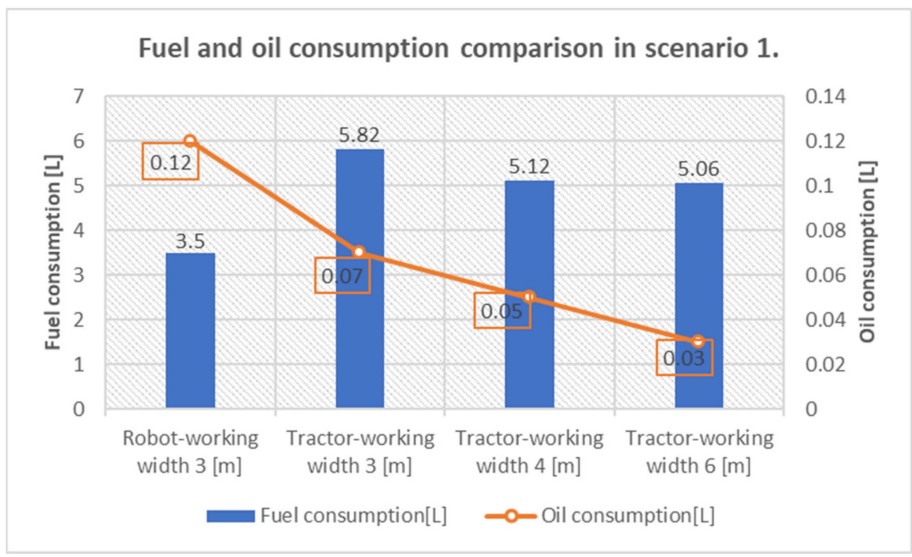

**Figure 11.** Comparison of fuel and oil consumption for robot and tractor in scenario 1.

Figure 11 shows that the robot has less fuel consumption than the tractor and it can save up to 39.8% of fuel till the end of the operation. The results also show that the robot has more oil consumption than the tractor.

Based on Figure 12, the comparison of operational costs shows that the robot has up to 60% less operational costs per hour than the tractor. A comparison of total costs shows that the robotic solution has up to 57% less total cost per hour than the tractor. Considering the duration of this operation, we can calculate the total cost of the operation. The results show that the tractor with 6-m working width has up to 46% less total cost than the robot. Moreover, the comparison of $CO_2$ emission reveals that the robot emits up to 39.8% less $CO_2$ than the tractor in this operation.

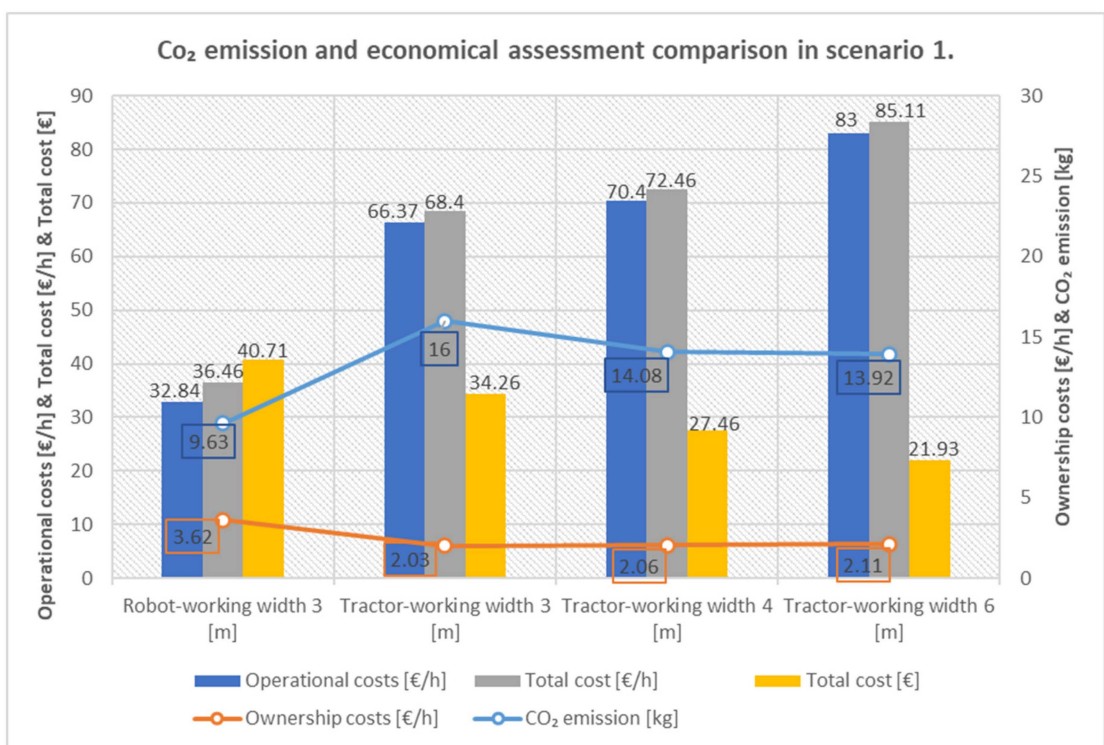

**Figure 12.** The comparison of operational and total cost and CO$_2$ emission.

In the second scenario, a sample field (Figure 13) with a size of fewer than 10 hectares was selected for seeding operation with the robot. The robot followed the initial plan generated by the route planning and the whole operation was accomplished automatically.

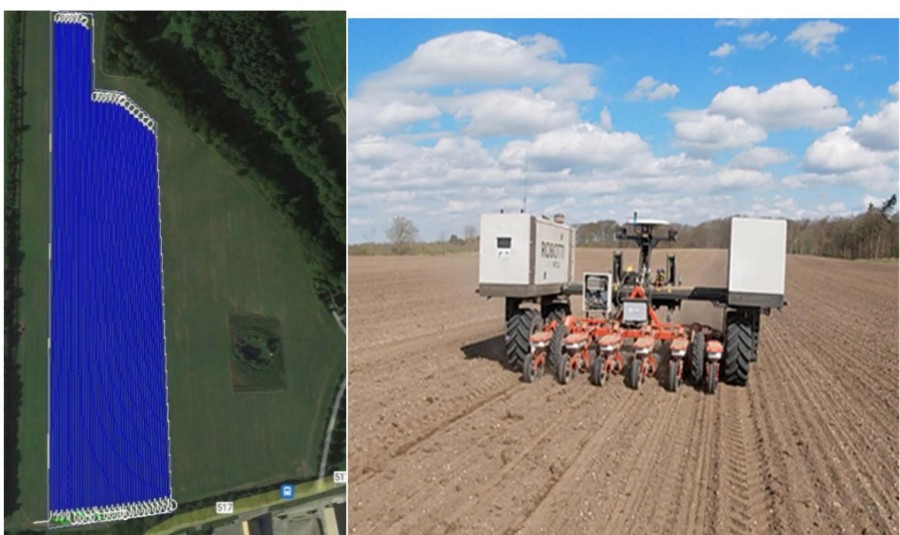

**Figure 13.** A sample field with a size equal to 7.5 (ha) was used for the seeding operation with the robot.

Table 9 shows the result of in-field operation for the robot and the tractor with different sizes of the implementation. The procedure for calculating the operational factors was the same as described in Table 8.

**Table 9.** The results of in-field operation for the second scenario for the robot and the tractor with different working widths.

| | Robot-Working Width 3 [m] | Tractor-Working Width 3 [m] | Tractor-Working Width 4 [m] | Tractor-Working Width 6 [m] |
|---|---|---|---|---|
| Field working time [min] | 167 | 167.2 | 126.1 | 83.6 |
| Non-working time [min] | 70 | 80.3 | 68.9 | 43.9 |
| Static time [min] | 9 | 12.75 | 4.5 | 4.5 |
| Total time [min] | 246 | 260.25 | 199.5 | 132 |
| Field efficiency [%] | 67.9 | 64.2 | 63.2 | 63 |
| Effective field capacity [ha/h] | 1.63 | 1.73 | 2.27 | 3.37 |
| Fuel consumption [L] | 43.5 | 52.28 | 46.06 | 44.6 |
| Oil consumption [L] | 0.46 | 0.59 | 0.45 | 0.3 |
| Energy cost [EUR] | 87 | 104.56 | 92.12 | 89.2 |
| Lubrication cost [EUR] | 0.138 | 0.177 | 0.135 | 0.09 |
| Labour cost [EUR] | 60.68 | 128.4 | 98.42 | 65.12 |
| Repair and Maintenance [EUR] | 47.97 | 58.56 | 45.55 | 30.89 |
| Operational costs [EUR] | 195.79 | 291.7 | 236.23 | 185.3 |
| Operational costs [EUR/h] | 47.75 | 67.25 | 71.05 | 84.23 |
| Ownership costs [EUR/h] | 3.62 | 2.03 | 2.06 | 2.11 |
| Total cost per hour [EUR/h] | 51.37 | 69.28 | 73.11 | 86.34 |
| Total cost [EUR] | 210.62 | 300.5 | 243.09 | 189.95 |
| $CO_2$ emission [kg] | 119.63 | 143.77 | 126.67 | 122.65 |

Figure 14 shows the comparison of field efficiency and field capacity between the robot and tractor in scenario 2. Based on the results, the robot has up to 7% better field efficiency than the tractor in this sample field. The comparison of effective field capacity shows that tractor has a higher field capacity than the robot.

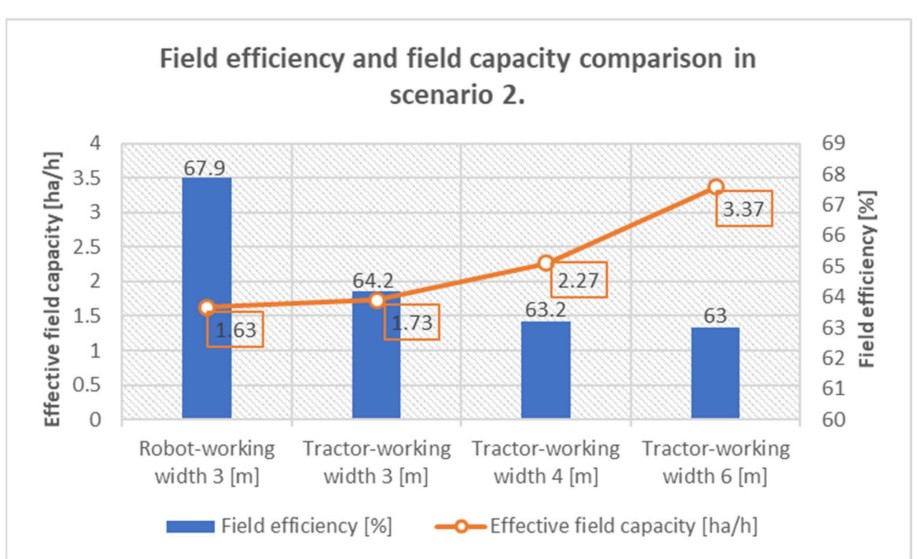

**Figure 14.** The comparison of field efficiency and field capacity for the robot and tractor in the second scenario.

According to Figure 15, the comparison of fuel consumption shows that the robot has up to 16.8% less fuel consumption than the tractor. Regarding oil consumption, the robot consumes less oil than the tractor with the same size of implementation. However, the tractor with 6 m size implementation, has less oil consumption than the robot.

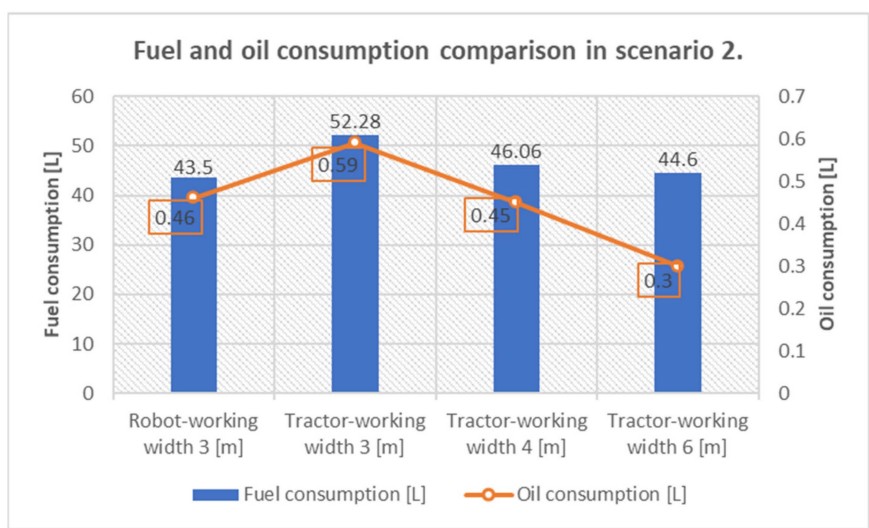

**Figure 15.** The comparison of fuel and oil consumption between robot and tractor in the second scenario.

Figure 16 demonstrates that in this sample field, the robot has up to 43.3% less operational costs per hour than the tractor with a different size of implementation. The comparison of total cost shows that the robot has up to 40% less total cost per hour than the tractor. The comparison of the total cost of operation shows that the robot has up to 30% less cost than the tractor with 3 and 4 m of working width. The total cost of operation for the tractor with 6-m working width is 10% less than the robot. Moreover, the comparison of $CO_2$ emission shows that the robot has up to 16.8% less $CO_2$ emission than the tractor in this operation.

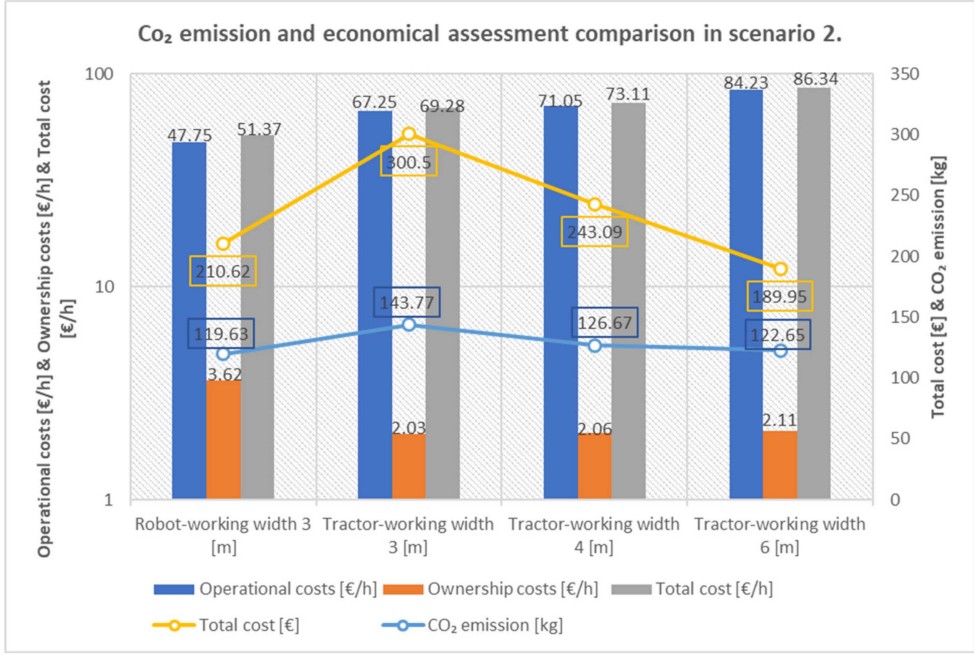

**Figure 16.** The comparison of economical assessment and $CO_2$ emission between robot and tractor in the second scenario.

In the third scenario, a sample field (Figure 17) with a size of more than 10 hectares was selected for seeding operation with the robot. The robot followed the initial plan generated by the route planning and all the operation was accomplished automatically.

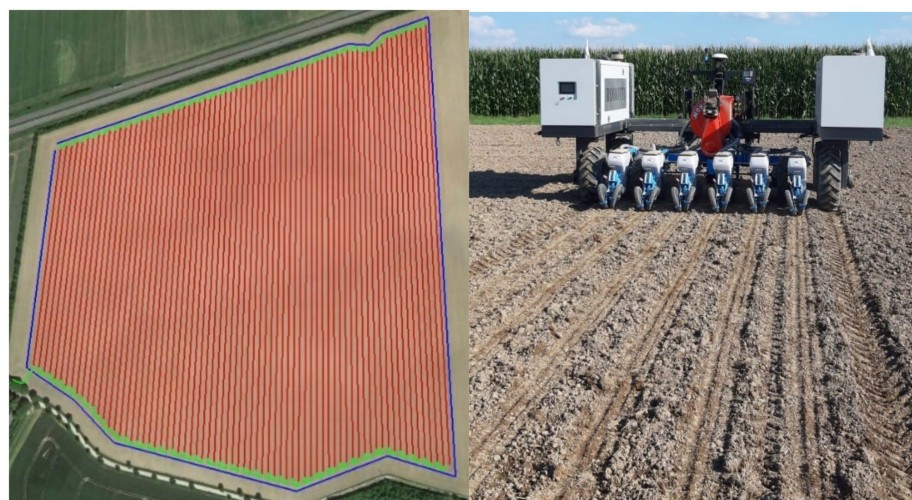

**Figure 17.** The sample field in the third scenario with a size equal to 21 ha.

Table 10 represents the result of in-field operation for the robot and the tractor with different sizes of implementation.

**Table 10.** The results of in-field operation for the third scenario for the robot and the tractor with different working widths.

|  | Robot-Working Width 3 [m] | Tractor-Working Width 3 [m] | Tractor-Working Width 4 [m] | Tractor-Working Width 6 [m] |
|---|---|---|---|---|
| Field working time [min] | 514.1 | 479.5 | 399.3 | 267.4 |
| Non-working time [min] | 89.7 | 117.4 | 101 | 69 |
| Static time [min] | 118.2 | 96.1 | 70.4 | 42 |
| Total time [min] | 722 | 693 | 571 | 378 |
| Field efficiency [%] | 71.2 | 69.2 | 69.9 | 70.7 |
| Effective field capacity [ha/h] | 1.71 | 1.87 | 2.52 | 3.82 |
| Fuel consumption [L] | 121.1 | 136.14 | 128 | 124.9 |
| Oil consumption [L] | 1.34 | 1.56 | 1.28 | 0.85 |
| Energy cost [EUR] | 242.2 | 272.28 | 256 | 249.8 |
| Lubrication cost [EUR] | 0.4 | 0.47 | 0.38 | 0.255 |
| Labour cost [EUR] | 178.1 | 341.9 | 281.7 | 186.48 |
| Repair and Maintenance [EUR] | 140.79 | 155.93 | 130.38 | 88.45 |
| Operational costs [EUR] | 561.5 | 770.6 | 668.5 | 525 |
| Operational costs [EUR/h] | 46.66 | 66.72 | 70.24 | 83.33 |
| Ownership costs [EUR/h] | 3.62 | 2.03 | 2.06 | 2.11 |
| Total cost per hour [EUR/h] | 50.28 | 68.75 | 72.3 | 85.44 |
| Total cost [EUR] | 605.04 | 794.06 | 688.06 | 538.27 |
| $CO_2$ emission [kg] | 333.03 | 374.38 | 352 | 343.48 |

Based on Figure 18, the comparison of field efficiencies shows that the robot has up to 3% better field efficiency than the tractor. The tractor has higher effective field capacity than the robot.

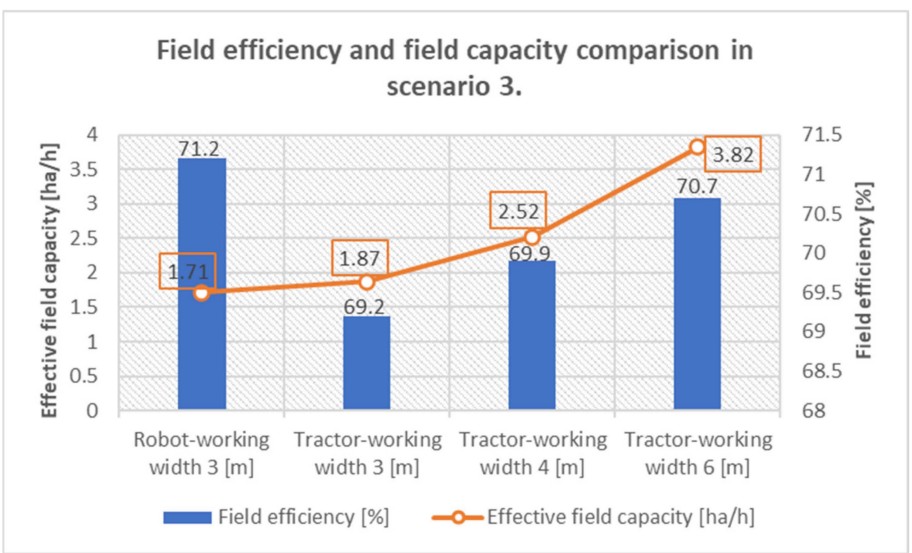

**Figure 18.** The comparison of field efficiency and field capacity for robot and tractor in scenario 3.

According to Figure 19, the robot has up to 11% less fuel consumption than the tractor. Regarding oil consumption, the robot consumes less oil than the tractor with the same size of implementation but, the tractor with 6 m size of implementation, shows less oil consumption than the robot.

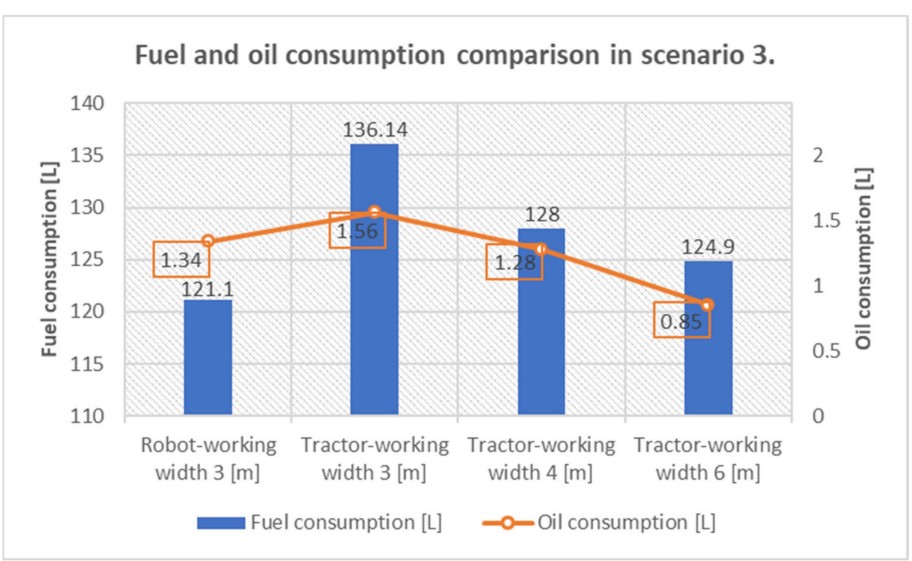

**Figure 19.** The comparison of fuel and oil consumption between robot and tractor in scenario 3.

Figure 20 shows that the robot has up to 44% less operational costs per hour than the tractor. The comparison of total cost shows that the robot has up to 41% less total cost per hour than the tractor. The comparison of the total cost of the operation shows that the robot has up to 28% less cost than the tractor with a 3–4 m working width. The tractor with 6 m working width has 11% less cost than the robot. Moreover, the comparison of $CO_2$ emission shows that the robot has up to 11% less $CO_2$ emission than the tractor.

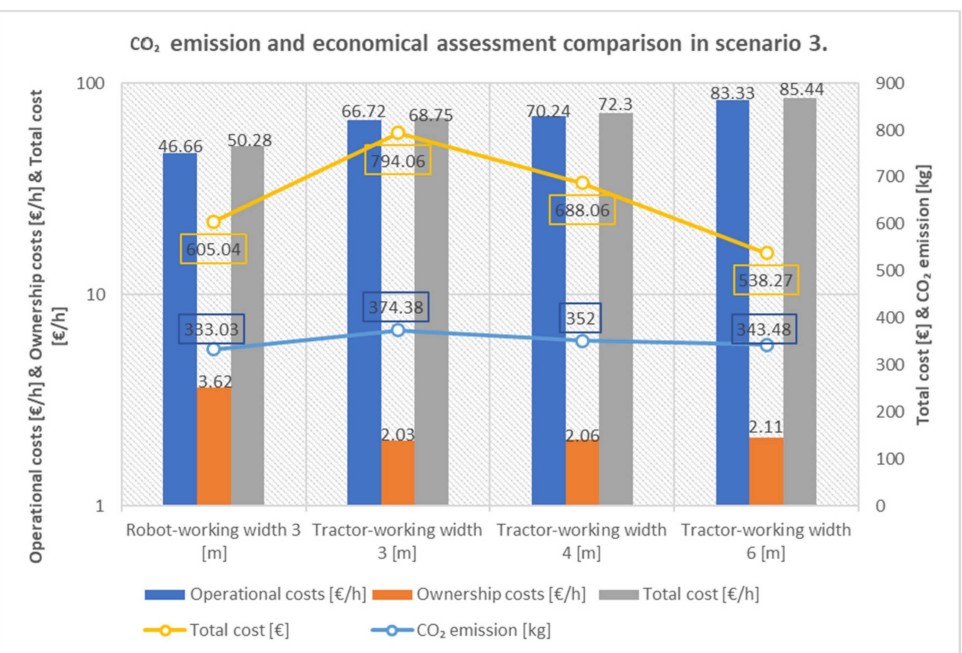

**Figure 20.** The comparison of economical assessment and $CO_2$ emission between robot and tractor in the third scenario.

### 3.3. Weeding Operation

In the fourth scenario, the sample field in the first scenario (Figure 21) with a size equal to 0.56 (ha) was selected for the weeding operation. The driving was connected to route planning and the operation was accomplished automatically and the duration of the operation was almost 76 min.

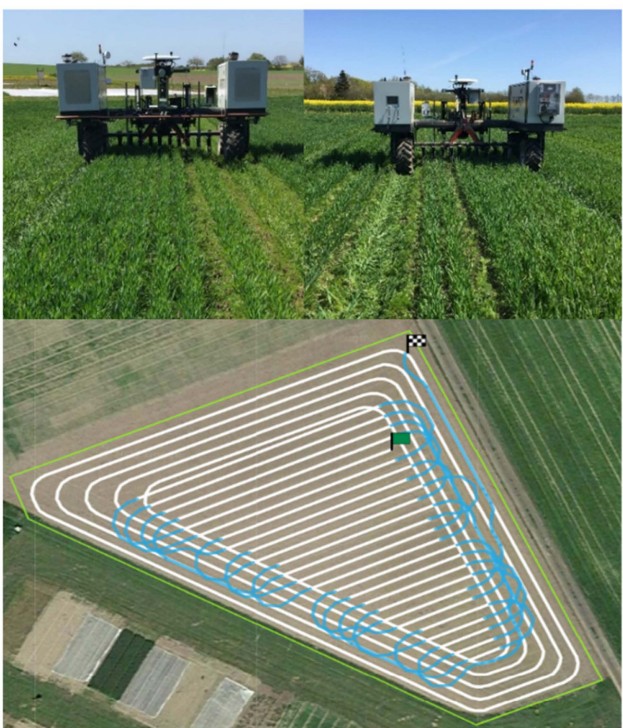

**Figure 21.** Weeding operation with the robot in a triangle shape field.

Table 11 represents the result of in-field operation for the robot and the tractor with different sizes of implementation. In this operation, the working speed of the robot was set

to 3.5 (km/h) and the travel speed of the robot was set to 5 (km/h). The working speed of the tractor was 5 (km/h) with the travel speed equal to 7 (km/h).

**Table 11.** The result of in-field operation for the robot and the tractor with different sizes of implementation in weeding operation.

| | Robot-Working Width 2.4 [m] | Tractor-Working Width 2.4 [m] | Tractor-Working Width 4 [m] | Tractor-Working Width 6 [m] |
|---|---|---|---|---|
| Field working time [min] | 43.23 | 26.5 | 14.8 | 9.98 |
| Non-working time [min] | 31.32 | 14.3 | 9.2 | 4.6 |
| Static time [min] | 1 | 6 | 4.65 | 4.65 |
| Total time [min] | 75.55 | 46.8 | 28.65 | 19.23 |
| Field efficiency [%] | 57.2 | 56.6 | 51.7 | 51.9 |
| Effective field capacity [ha/h] | 0.48 | 0.67 | 1.03 | 1.56 |
| Fuel consumption [L] | 2.7 | 7.36 | 4.5 | 3.02 |
| Oil consumption [L] | 0.14 | 0.11 | 0.06 | 0.04 |
| Energy cost [EUR] | 5.4 | 14.72 | 9 | 6.04 |
| Lubrication cost [EUR] | 0.04 | 0.03 | 0.02 | 0.01 |
| Labour cost [EUR] | 18.64 | 23.1 | 14.13 | 9.5 |
| Repair and Maintenance [EUR] | 14.73 | 10.5 | 6.6 | 4.6 |
| Operational costs [EUR] | 38.81 | 48.35 | 29.75 | 20.15 |
| Operational costs [EUR/h] | 30.82 | 62 | 62.3 | 62.9 |
| Ownership costs [EUR/h] | 3.62 | 2.03 | 2.08 | 2.14 |
| Total cost per hour [EUR/h] | 34.44 | 64.03 | 64.38 | 65.04 |
| Total cost [EUR] | 43.37 | 49.94 | 30.74 | 20.85 |
| $CO_2$ emission [kg] | 7.43 | 20.24 | 12.37 | 8.31 |

Figure 22 represents the comparison of operational factors between the robot and the tractor. The comparison of field efficiency shows that the robot has up to 9.6% better field efficiency than the tractor. However, the tractor has a higher effective field capacity than the robot.

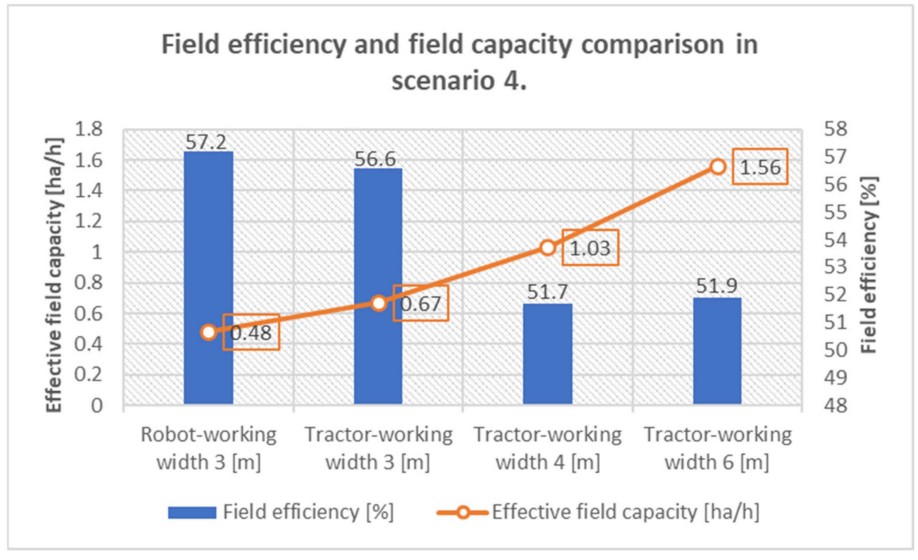

**Figure 22.** The comparison of field efficiency and field capacity between the robot and tractor in scenario 4.

Figure 23 shows the comparison of fuel and oil consumption between robot and tractor. The comparison shows that the robot has up to 63.3% less consumption than the tractor with different working widths of implementation. Moreover, the robot consumes more oil than the tractor in this case.

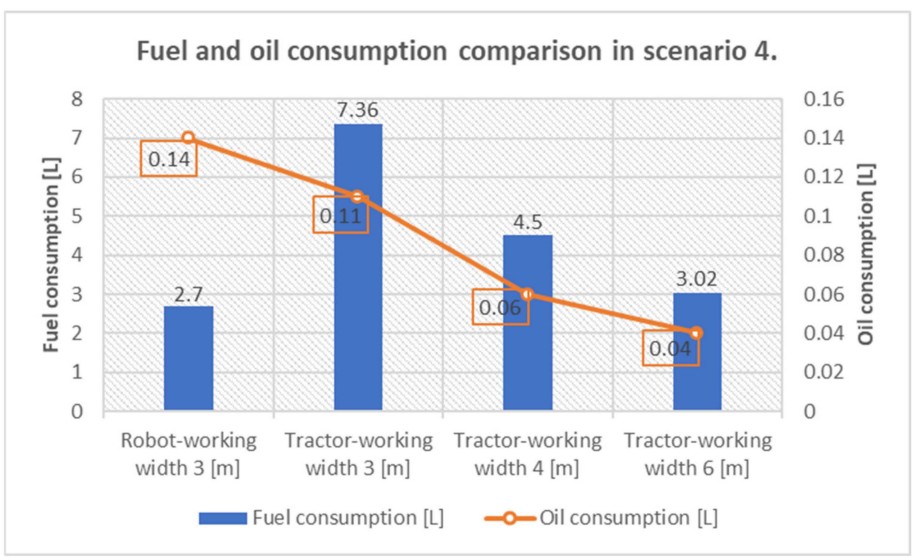

**Figure 23.** The comparison of fuel and oil consumption for the robot and the tractor in weeding operation.

Based on Figure 24, the comparison of operational costs shows that the robot has up to 51% less operational cost per hour than the tractor. A comparison of total costs per hour shows that the robot has up to 47% less total cost per hour than the tractor. The results shows that the tractor with 6 m working width has the lowest total cost. Moreover, the comparison of $CO_2$ emission shows that the robot has up to 63% less $CO_2$ emission than the tractor in weeding operation.

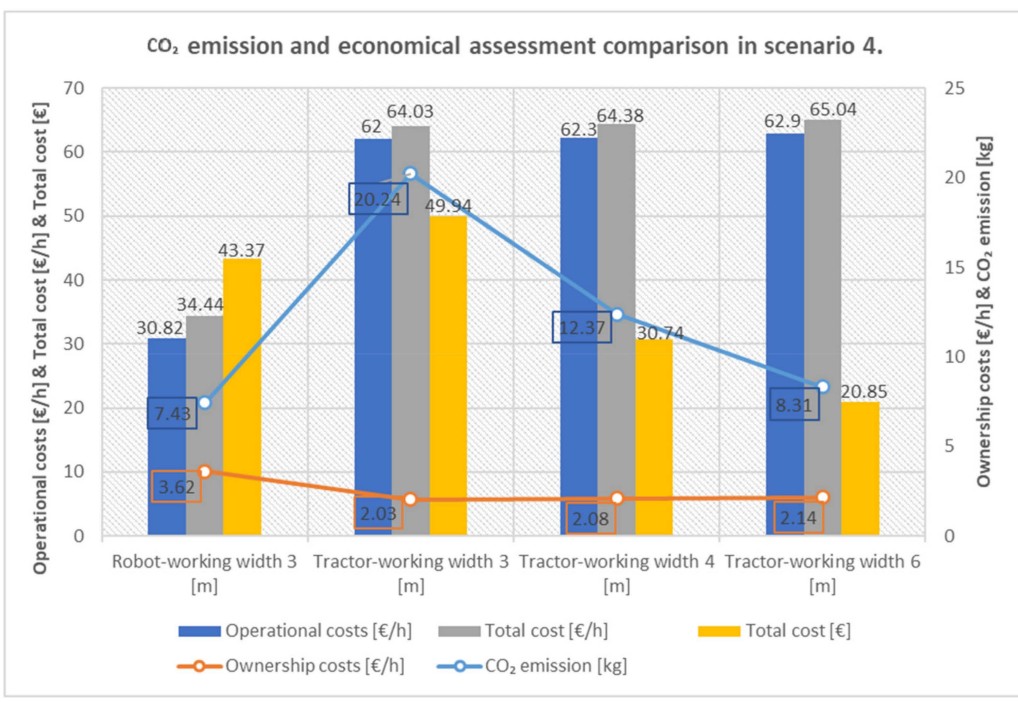

**Figure 24.** The comparison of economical assessment and $CO_2$ emission between robot and tractor.

The results show that in all the scenarios, the robot has less effective field capacity than the tractor since the tractor has a higher working speed and higher working width. The field speed of the tractor in seeding operation is up to 11 (km/h) and for the robot, it is up to 8 (km/h). Therefore, the tractor can accomplish the operation faster than the robot. The theoretical field capacity of this robot is 2.4 (ha/h), however, the working speed of the robot is varying based on the type of operation and the condition of the soil.

The shape of the agricultural fields is another important factor that can affect the operation of the robot. The robot has better field efficiency in the fields with long working areas (tracks) such as the sample fields in scenarios 2 and 3.

The size of the field is another factor that can affect the operation of the robot. The results show that by increasing the size of the field the efficiency of the robot is going to increase. One advantage of robots is that they can work continuously for a longer time and this benefit of robots can better show in a large field where the duration of the operation is longer. In the third scenario where the operation takes a longer time, we consider 40 min as lunchbreak for the operator of the tractor. Table 12 shows the result of this change in the operational and economic factors.

**Table 12.** The results of in-field operation in the third scenario for the robot and the tractor by considering 40 min break time for the tractor.

| | Robot-Working Width 3 [m] | Tractor-Working Width 3 [m] | Tractor-Working Width 4 [m] | Tractor-Working Width 6 [m] |
|---|---|---|---|---|
| Field working time [min] | 514.1 | 479.5 | 399.3 | 267.4 |
| Non-working time [min] | 89.7 | 117.4 | 101 | 69 |
| Static time [min] | 118.2 | 96.1 | 70.4 | 42 |
| Idle time [min] | - | 40 | 40 | 40 |
| Total time [min] | 722 | 733 | 611 | 418 |
| Field efficiency [%] | 71.2 | 65.4 | 65.3 | 64 |
| Effective field capacity [ha/h] | 1.71 | 1.77 | 2.35 | 3.46 |
| Fuel consumption [L] | 121.1 | 136.14 | 128 | 124.9 |
| Oil consumption [L] | 1.34 | 1.56 | 1.28 | 0.85 |
| Energy cost [EUR] | 242.2 | 272.28 | 256 | 249.8 |
| Lubrication cost [EUR] | 0.4 | 0.47 | 0.38 | 0.255 |
| Labour cost [EUR] | 178.1 | 349.4 | 291.2 | 199.25 |
| Repair and Maintenance EUR] | 140.79 | 164.93 | 139.5 | 97.8 |
| Operational costs [EUR] | 561.5 | 787.1 | 687.1 | 547.11 |
| Operational costs [EUR/h] | 46.66 | 64.43 | 67.47 | 78.53 |
| Ownership costs [EUR/h] | 3.62 | 2.03 | 2.06 | 2.11 |
| Total cost per hour [EUR/h] | 50.28 | 66.46 | 69.53 | 80.64 |
| Total cost [EUR] | 604.2 | 811.92 | 708.05 | 561.8 |
| $CO_2$ emission [kg] | 333.03 | 374.38 | 352 | 343.48 |

The results show that the field efficiency and field capacity for tractor with working widths 3, 4, and 6 m were reduced by 5%, 6%, and 9% respectively. The labour cost, repair and maintenance cost, operational cost, and the total cost of the seeding operation were increased by up to 9%.

The weight of agricultural machinery as an important factor can affect the amount of compaction on the soil. The weight of this robot is 3100 (kg), and the tractor (John Deere 7R250) is 12,058 (kg). The comparison shows that the weight of the tractor is almost four times higher than the robot. The higher weight of implementation can also put more stress on the topsoil layer which can cause more soil compaction [34,35].

One of the benefits of agricultural robots is that they can operate instead of a human in a difficult situation and accomplish repetitive tasks. For example, when a lot of hand weeding is required, this job can put too much effort and stress on workers. Applying

robotic solutions in these types of field tasks can reduce the total operational time as well as being able to reduce the workload of workers and increase the leisure time on family farms [8,36].

About the price of the tractor, there are some factors that can affect the price of a tractor such as engine power, size, brand, and so on. For this research, we needed a tractor with at least 150 kW power to be able to carry out all the operations. There is a wide range of prices for this type of tractor in the market. To fairly show the effect of the tractor's price on the ownership costs, we consider a range of EUR 110,000–270,000 for the purchase price of the tractor. Table 13 shows the calculation of ownership costs for a cheaper tractor in both seeding and weeding operations.

**Table 13.** The calculation of ownership costs for the tractor in seeding and weeding operations considering the lowest purchase price for the tractor from the mentioned range.

| Seeding Operation | | | | |
| --- | --- | --- | --- | --- |
| | | **Tractor (3 m)** | **Tractor (4 m)** | **Tractor (6 m)** |
| Ownership costs parameters | Purchase price (EUR) | 114,500 | 118,500 | 126,000 |
| | Salvage value (EUR) | 11,450 | 11,850 | 12,600 |
| | Economical life (years) | 15 | 15 | 15 |
| | Average interest rate (%) | 9 | 9 | 9 |
| | Depreciation (EUR/year) | 6870 | 7110 | 7560 |
| | Interest (EUR/year) | 4637 | 4799 | 5103 |
| | Insurance & Taxes (EUR/year) | 1145 | 1185 | 1260 |
| | Housing (EUR/year) | 1145 | 1185 | 1260 |
| Ownership costs (EUR/year) | | 13,797 | 14,279 | 15,183 |
| Economical life (h) | | 16,000 | 16,000 | 16,000 |
| Ownership costs (€/h) | | 0.86 | 0.89 | 0.95 |
| Weeding operation | | | | |
| | | Tractor (2.4 m) | Tractor (4 m) | Tractor (6 m) |
| Ownership costs Parameters | Purchase price (EUR) | 114,200 | 121,000 | 129,900 |
| | Salvage value (EUR) | 11,420 | 12,100 | 12,990 |
| | Economical life (years) | 15 | 15 | 15 |
| | Average interest rate (%) | 9 | 9 | 9 |
| | Depreciation (EUR/year) | 6852 | 7260 | 7794 |
| | Interest (EUR/year) | 4625 | 4900 | 5261 |
| | Insurance & Taxes (EUR/year) | 1142 | 1210 | 1299 |
| | Housing (EUR/year) | 1142 | 1210 | 1299 |
| Ownership costs (EUR/year) | | 13,761 | 14,580 | 15,653 |
| Economical life (h) | | 16,000 | 16,000 | 16,000 |
| Ownership costs (EUR/h) | | 0.86 | 0.91 | 0.98 |

The results show that considering a cheap price tractor can reduce the ownership costs in both seeding and weeding operations by up to 57%. However, the ownership costs take a small part of the total cost of the operation.

Agricultural robots can affect the labor market as well. For example, they can take the place of low-skilled labour in the field to save on wage and transaction costs [2] or may also allow replacing skilled labor with unskilled labor because of new automation technology in steering systems which enables driving tractor autonomously without need to a skillful operator. Moreover, agricultural robots require robot software and sensor experts which underlines the demand for more high-skilled labor due to digitalization in agriculture [37].

Considering all the positive aspects of the robotic solutions, there are some concerns about these autonomous systems such as unresolved liability issues, market maturity and availability of robots, increased dependence on the manufacturer, lack of compatibility with other technology, high level of specialization of technology, the safety of humans and animals and ethical concerns [38].

## 4. Conclusions

In this paper, the operational, economic, and environmental analysis of an agricultural robotic system was presented and some of the important operational indicators as well as individual cost elements with some environmental factors were provided. Additionally, a case study of a robotic system performing seeding and weeding operations in three different fields was presented and the calculated operational, economic, and environmental factors were compared to the factors of the respective conventional system. Based on the results, the robotic system has 3–9.6% better operational efficiency than conventional machinery. However, the comparison of effective field capacity shows that the conventional system has a 2–3.6 times bigger field capacity than the robotic system. The difference in the field capacity is due to the larger working width of the tractor's implementations and the higher operating speed of the tractor. Moreover, the total cost per hour of the operation executed by a robotic system in all the scenarios ranged between 40–57% less than the total cost per hour of the operation executed by conventional agricultural machinery. The tractor with 6-m working width has up to 46% less total cost than the robot due to the higher field capacity. The comparison of fuel consumption between the robotic system and the conventional one shows that the robotic system can save 11–63.3% of diesel during the entire operation. The labor cost highly affects the cost of the robotic system due to the longest duration of the operations and removing human workers from the robotic system can make it more economic, however, completely removing humans from the robotic system is not currently possible. Furthermore, the comparison of the amount of emission $CO_2$ gas during the operation between robotic and conventional systems shows that the robotic system can emit 11–63.3% less $CO_2$ gas into the environment. Finally, a comparison of the weight between the robotic and conventional systems shows that the robotic system is four times lighter than the conventional system and it has fewer effects on the soil structures in comparison with the heavy conventional machinery. In a future study, the social effects of these new robotic systems on the farmers and workers in the agricultural field are going to be considered.

**Author Contributions:** Conceptualization, M.V., R.G. and C.A.G.S.; methodology, M.V. and R.G.; software, M.V.; validation, M.V., R.G. and C.A.G.S.; formal analysis, M.V.; investigation, M.V.; resources, R.G. and C.A.G.S.; data curation, M.V.; writing—original draft preparation, M.V.; writing—review and editing, M.V., R.G. and C.A.G.S.; visualization, M.V.; supervision, R.G. and C.A.G.S.; project administration, C.A.G.S.; funding acquisition C.A.G.S. All authors have read and agreed to the published version of the manuscript.

**Funding:** This research was funded by Innovation Fund Denmark, Grant 9092-00007B AgroRobottiFleet.

**Conflicts of Interest:** Authors declare no conflict of interest.

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
