# Peer review of "Operational, Economic, and Environmental Assessment of an Agricultural Robot in Seeding and Weeding Operations"

_agriengineering, doi:10.3390/agriengineering5010020_

Round 1

Reviewer 1 Report

I am an agronomist, not an engineer.  So, this will likely be obvious in my comments.  

Overall, I thought this paper did a nice job comparing the robotic system to the conventional system.  I liked the fact that numerous field size scenarios were reported and for two different field operations.

Suggested edits are below:

1.  Line 54:  the sentence is not clear.  It seems that a word is missing following “….. soil workability ….”

2.  Was there any agronomic differences in stand establishment?  Was there swath control system on the wider seeder?  If not, there could be additional seed cost saving and higher yields in the narrower (3 m) machines. overlapping in planting leading to higher seeding costs?

 3.  One advantage of robots is that they should not need breaks, to stop for meals, and can have longer work days.  I like the fact that the information was presented on a field basis, but it would be interesting to calculate the efficiency, following current labor laws.  I understand that all factors cannot be discussed in this manuscript, but it would add value to the manuscript is a scenario or two was described.    

4.  On figures 10,11, 13, 14, 16, 17, 19, 20, I would prefer these data presented differently.  The stacked bar charts, when looking at percentages, is not the most effective way to present the data.  

5.  Line 468-472: there were not data presented on soil compaction comparing the two systems.  There are plenty of prior publications on the impact of weight on compaction.  If this is going to be a major result and conclusion, then the assumptions of estimating compaction for the different systems should be clearly stated in the manuscript.  If the importance of weight on soil compaction is just being emphasize as an additional benefit to the robotic system, then citations should be added to this paragraph.     

6.  Line: 566, no author(s) is listed for this reference

7.  Line 570:  Citation 19 is formatted differently than others

8.  530-590: some citations end with a period and others do not.  I am unfamiliar with this journal.  So, maybe this is correct formatting.

Reviewer 2 Report

I have read the paper "Operational, economic, and environmental assessment of an agricultural robot" that compares the use of a robot and a tractor for certain agricultural tasks.

The paper addresses a relevant aspect of field robots, their impacts.

General remarks.

You are not consistens in the use of units on the provided physical quantities.

Sometimes they appear in parenthesis (eg. L118) whereas in other locations they appear without parenthesis (e.g. L114).

What is the most relevant cost, the total cost for performing the operation or the cost per hour for completing the operation?

I would argue that it is the "total cost" that is of interest, but you have chosen to focus on the "total cost per hour".

The figures that present the performance of the different types of equipment should be improved. Please include the scenario number / description in the figure. I don't think that a stacked bar plot is suitable for visualizing these data. See an alternative type of visualization in the attached file. Also show the "Total cost [€]" as a performance parameter in the plots.

Specific comments:

L22: The numbers on saved fuel and CO2 reductions are identical. They might be described in one sentence instead of two separate sentences.

L72: Formatting of the reference "Sunitha K. A et al (2017)" -> "Sunitha et al (2017)".

L117: What do you mean by zero turn?

L148: Be consistent with the units for hours: km/h -> km/hr

L170: Please be aware that you have two different values which are both denoted SV (scap value). I assume that the scap value of the old tractor could be different that the expected scap value of the new tractor. Please note that in the text.

L214: Units: "KW.hr/ha" -> "kW hr / ha"

L252: Is the robot allowed to move at 12 km / hr as description in the image caption?

L253: Can you say anything about the soil type of the test field.

L262-266: You seems to be converting to and from electrical horse powers (by multiplying / dividing with 1.3405). Why not keep the numbers in SI units?

L276: Figure 7, please make it easier to read the axis labels and the numbers on the axes.

L285: Equation 7. I assume that the notation $B(S)$ means that B should be multiplied with S. Formally this notation means that B is a function of S. I suggest to change it to the following. $F_i (A + B \cdot C + C \cdot S^2) W T$.

L288: I don't understand the "1 (dimensionless) for minor ..." part.

L294: Where do the ranges 1.1 - 39 and 1.5-4.8 come from?

L307: You assume that the robot only works for 600 hours each year, whereas the tractor could be used for 1067 hours each year.  Why this difference in utilization?

L360: I cannot find a description of the "Static time [min]" row in table 7.

L368: The visualisation of data can be improved. Please see the attached example of a different visualisation of the shown data.

L371 and L372: Include "per hour" when the operational cost is discussed.

L473-L478: I find this paragraph hard to read. 

L502-L504: I find these three lines extremelyl misleading. You are mixing the values for "total cost" and the "total cost per hour". As the robot uses more time on the same task, it has a lower "total cost per time". I would argue that the interesting parameter to compare is the "total cost", in which case the robot is the most expensive on the first scenario (36.67 € vs 33.11 € - 21.17 €). 

L516: "In the future study " -> "In a future study"

L525: Remove the "

Reviewer 3 Report

1.     Initial Submission

1.1  Recommendation

Major Revision

2.     Comments to the Author

Ms. Ref. No.: agriengineering-2059238

Title: Operational, economic, and environmental assessment of an agricultural robot. Case seeding/weeding operations

Authors: Mahdi Vahdanjoo, René Gislum and Claus Aage Grøn Sørensen

Overview and general recommendation:

The manuscript is targeting a very important aspect for the uptake of robotic solutions in agriculture and is of high relevance to the sector. The methodology is clearly structured and the results are presented in a clear and comprehensive way. However, there are still some aspects that need improvement. Thus, my recommendation is that the manuscript requires a Major Revision. You may find my comments below, referenced for the specific locations within the manuscript.

Comments

-       Abstract: You need to rephrase the abstract as to not be text identical with other sections of the manuscript.

-       Section 2.1: It would be good to mention why you chose the specific robot over others. Also reference the technical specs and capabilities from the manufacturer.

-       Figure 2: Please explain how the information inside was produced (what does automatic not working means for example and why on these regions).

-       Line 315: You do not mention why did you chose the John Deere option as well, as it is one of the pricier in the market. In addition, a price range for machinery capable of performing the tasks described, both conventional and robots, would be much fitting since you are performing an economic assessment.

-       Conclusions: The results, even though calculated very accurately, do not compensate for the fact that there are far cheaper conventional tractor options other than the one picked for this study. A good practice would be to consider the cheapest capable option that can cover these tasks, as well as one of the pricier, such as the John Deere one, and then offer a percentage variance in performance benefits or deficits. Also, an explanation on why a conventional system of similar cost was not chosen would be required. This is a major aspect, since it renders your research heavily biased towards the robot.

Reviewer 4 Report

Reviewer’s comments on Manuscript No. agriengineering-2059238 

Title

:

Modification Needed. Marge both sentences

Abstract

:

Need modifications and grammatical corrections.

A brief description of robot must be added to this section. Put More results of the study.

Keywords

:

Need modifications. Arrange in Alphabetical Order

Introduction

:

An introduction of a research paper should have the following in sequential order to make the paper acceptable.

1.         The basic idea of the topic of the paper

2.         Citation of similar works done by others quoting the references in proper format of the journal.

3.         Some modifications as suggestion are as follow:

Line 37 – 39 break the statement into two sentences

Line 41-41 Rephrase the statement.

Line 37-56 elaborate more on types of different agricultural robot on the basis of power source its merits and demerits. Reasons of selection of power source of your robot.

Line 53-55 Rephrase the statement.

Line 56 would reduce the size, initial cost of machinery and operation cost.

Line 57-74 the knowledge gap in all recent research must be placed in continuous statement regardless of the researchers/year

Line 75-88 Same as above

Line 101-106 Hypothesis missing

Materials and Methods

:

Need modifications as following suggestions:

1.      For proper and authentic results there should be proper and authentic as well as standard method

2.      there is no citation in the methodology, it shows that it has been invented by you.

3.      Use citation to denote the methods. Be precise and focused and use Standard SI Units. Abbreviation must pe explained.

4.      Statistical model should be clearly discussed

5.      No Design of Experiment was discussed

6.      Subtitles in 2.3 write in correct order

7.      Equations number not mentioned in main text

8.      In subheading 2.4 Partial factors was discussed Net Return, Gross Profit, B:C ratio and Payback Time also need to describe & observe.

9.      Critical citation missing at line 156-157, 159-161, 162-163, 194-195, 195-196, 196-197, 222-227.

10.  Line 237-241 marge both paragraph as well as Figures are not mentioned in main text.

11.  In Table 3 and Table 4 give Mean values of each parameters as well as put this in results section

12.  Figure 7 and Figure 8 put it in Results and increase the font size of the markers used in figures for be readable.

13.  Why only one width is selected for robot and multiple for tractor operation.

14.  For operation of robot highly skilled labour is required so why the labour cost become half as compared to tractor operation.

Result and Discussion

:

Need modifications as suggested

1.      The manuscript lacks in proper results and discussion

2.      The discussion is explanation of your results in the light of similar works done by citing them to make your statement valid.

3.      Revise this portion to make your paper relevant. Some more recent references should be incorporated to strengthen the advocated facts.

4.      Line – 333 a size of 0.56 ha (less then 1 ha) was selected for seeding

Line 344-345 how the oil consumption was calculated or measured not explained in materials and methods

Line 359 Not discussed in material and method and table not mentioned in main text.

In table 7 fuel consumption is very less but the power of engine used in robot was just double of that used in tractor, How??

The ownership cost only contains the robot/tractor cost not of the implement used which was actually varying during study. Must be added in this therefore whole economics become invalid. Same was also found in table 8, table 9 and table 10.

Line 364-367 Two contradictory statements. Must be supported with valid reason and citation

Line 392 use higher in place of bigger

Line 395 its 4 m not 6 m check with data also in line 421

Line 441-444 also the operating speed was different both robot and tractor why??

Line 471-472 No comparison was done in main study then where did this statement come in results

Conclusion

:

Need modifications as suggested above section of manuscript with following suggestions

1.      The manuscript includes conclusion of that study which is not carried out in the paper, modify it.

2.      Manuscript should include the conclusion based on the results of the study shown.

Overall Comments:

1.        The paper may please be accepted with major revision as suggested.

Round 2

Reviewer 2 Report

L337: Figure 7. It is still very hard to read the tick marks on the x axis. 

L346: Figure 8. It is still very hard to read the tick marks on the x axis. Please also add a label to the y axis that specifies that the numbers are in kilo Newton.

L348: Please use proper SI prefixes. Kilowatt is kW and kilonewton is kN.

L386: Figure 10: You could consider to use two separate x axes, so both charts (Effective field capacity and Field efficiency) could use most of the available space. Right now the effective field capacity is shown on a small part of the figure. An alternative would be to scale the values relative to the values from the robot, such that the robot would have a value of 100 %. Here the raw value could still be shown next to the bar that visualises the percentage value.

The comment to figure 10 also applies to the rest of similar figures in the paper. Ie. figure 11, 12, 14, 15, 16, 18, 19, 20, 21, 22 and 23.

Reviewer 3 Report

All the previous comments were addressed adequately.

Author Response

Thank you.

Reviewer 4 Report

Replace CO2 with CO2 in whole manuscript.

Replace m2 with m2

Replace KW hr with kW hr

Replace KW hr/ha with kW hr/ha

Replace symbol of liter (L) to “l”

Replace KW with kW

Line 54-65 - add References

All previous suggestions were incorporated in manuscript. Kindly also do these minor revisions.

Best of luck for publication.

Author Response

The authors appreciate the reviewer for pointing out these issues. All the mentioned points were considered in the manuscript.
